# Integrative Transcriptomic and Metabolomic Analysis Provides New Insights into the Multifunctional ARGONAUTE 1 Through an *Arabidopsis ago1-38* Mutant with Pleiotropic Growth Defects

**DOI:** 10.3390/plants15010044

**Published:** 2025-12-23

**Authors:** Xiangze Chen, Xinwen Qing, Xiaoli Peng, Xintong Xu, Beixin Mo, Yongbing Ren

**Affiliations:** Guangdong Provincial Key Laboratory for Plant Epigenetics, College of Life Sciences and Oceanography, Shenzhen University, Shenzhen 518060, China; chenxz@szu.edu.cn (X.C.); coratheone@163.com (X.Q.); pengxiaoli@szu.edu.cn (X.P.); xxt19950625@163.com (X.X.)

**Keywords:** transcriptome, metabolome, *ago1-38*, siRNA processing, phytohormone signal, plant–pathogen interaction

## Abstract

ARGONAUTE 1 (AGO1) selectively recruits microRNAs (miRNAs) and some small interfering RNAs (siRNAs) to form an RNA-induced silencing complex (RISC) to regulate gene expressions and also promotes the transcription of certain genes through direct chromatin binding. Complete dysfunction of AGO1 causes extremely serious growth arrest and sterility in Arabidopsis. Here, we characterize an *ago1-38* allele with distinctive morphological abnormalities obviously distinguishing it from the other *ago1* alleles, such as *ago1-25* and *ago1-45*. The aberrant phenotypes of *ago1-38* were completely restored in its transgenic complementation lines harboring an AGO1 promoter and coding sequence. To investigate the mechanism underlying the unique phenotype of *ago1-38*, integrated transcriptomic and metabolomic analysis was employed. The glutathione metabolism pathway was significantly co-enriched in the integrated analysis of *ago1-38*, suggesting an altered balance of the glutathione-related redox system. Transcriptomic analysis showed that many genes in the siRNA processing pathway were significantly changed in *ago1-38*, suggesting the dysregulation of the siRNA pathway. Meanwhile, numerous genes, particularly the large set of transcriptional factors associated with plant–pathogen interaction networks and phytohormone signaling cascades, exhibited altered expression patterns, implying perturbed immune defense and hormonal signaling. Collectively, these findings provide new insights into the multifaceted roles of AGO1 in siRNA processing, pathogen response, and phytohormone signaling.

## 1. Introduction

Small RNAs (sRNAs), typically 21–24 nucleotide (nt) in length, are essential regulators of gene expression [1]. They play key roles in the modulation of plant growth, genome stability, and stress response [2]. Canonical plant sRNAs are mainly classified into microRNAs (miRNAs) and small interfering RNAs (siRNAs), both of which are processed from longer RNA precursors by Dicer-like (DCL) endoribonucleases [2]. These precursors are either hairpin-like structures processed from the folding of self-complementary RNAs or double-stranded RNAs (dsRNAs) synthesized by RNA-dependent RNA polymerases (RDRs) [2]. Generally, miRNAs are produced from hairpin-like precursors, whereas siRNAs are generated from RDRs-dependent dsRNAs. Mature sRNAs are subsequently loaded onto AGO proteins to form RISCs, which target endogenous or exogenous RNAs based on sequence complementarity and mediate gene or transposon silencing [3,4].

*Arabidopsis genome* encodes 10 AGO paralogues by 10 *AGO* genes, each participating in distinct small RNA pathways [5,6,7]. Among them, AGO1 is the best characterized and the most essential AGO protein in plants [8], which loads the majority of miRNAs and a subset of siRNAs, including virus-derived siRNAs to direct post-transcriptional gene regulation and antiviral defense [5,9]. Beyond its canonical sRNA silencing function, AGO1 also binds to the chromatin of specific actively transcribed genes and promotes their expression [10,11]. A series of *ago1* mutants have been screened and different alleles exhibit different growth defects in varying degrees [5], and a null mutant of *AGO1* (such as *ago1-24*) displayed extremely serious growth arrest [12]. The diverse and pleiotropic phenotypes of different *ago1* alleles suggest the multiple functions of AGO1 in plant growth and development.

Multi-omics technologies enable comprehensive and high-throughput analysis of biological samples. The combination of the transcriptome and metabolome is the most frequently used multi-omics strategy in plant research. Transcriptomics reveal gene functions and expression patterns under various conditions, whereas metabolome information connects these transcriptional changes to alterations in metabolites involved in multiple pathways, reflecting the downstream outcomes of corresponding gene expression [13].

Phytohormones act as chemical messengers that transfer information between cells [14] and influence nearly all aspects of plant development, from pattern formation during development to responses to biotic and abiotic stress [15]. Abundant physiological data suggest that plant hormones exhibit substantial cross regulation, with their signaling pathways interacting at the level of gene expression [15]. Plant–pathogen interactions involve intricate signaling networks that coordinate the plant immune response, in which phytohormone signaling plays a pivotal regulatory role [16]. In parallel, small interfering RNA (siRNA) functions as the plant’s main “molecular weapon” in defense against RNA viruses [17]. Hence, pathways associated with siRNA processing, phytohormone signaling, and plant–hormone interactions are often closely interconnected in plants.

In this study, we focused on the *ago1-38* allele, which exhibits a distinctive and pleiotropic phenotype in comparison to other *ago1* alleles, and performed integrated deep transcriptomic and widely targeted metabolomic analysis. The *ago1-38* allele displayed ultra-twisted and coiled leaves, slender petioles, short and little siliques, small leaves regenerated from big ones, and poor fertility. Both the transcriptomic and metabolomic profiling of *ago1-38* are largely distinct from those of *ago1-25* and *ago1-45*. The glutathione metabolism pathway was significantly co-enriched in the integrated transcriptomic and metabolomic analysis of *ago1-38*. Moreover, the expression of genes involved in the siRNA processing pathway, such as components of RNA polymerase Ⅵ (PolⅥ) and Ⅴ (PolⅤ), RDRs, suppressor of gene silencing (SGS), and players in the RNA-directed DNA methylation (RdDM) pathway, was significantly disturbed in *ago1-38*. In addition, the expression of massive genes (including a large set of transcriptional factors) associated with plant–pathogen interaction networks and phytochrome signaling cascades was also dramatically changed. These findings provide new insights into the multifaceted roles of AGO1 in the modulation of the siRNA processing pathway, plant–pathogen interaction networks, and phytohormone signaling cascades.

## 2. Results

### 2.1. The Peculiar Phenotype of the ago1-38 Allele

*AGO1* encodes an RNA slicer that selectively recruits the majority of miRNAs and some small interfering RNAs siRNAs to form an RISC, where it acts as an endonuclease to cleave target mRNAs. Null mutants of *AGO1* cause extremely serious plant growth inhibition and sterility. Until now, a series of *ago1* mutants with weakened AGO1 activities have been described and used for AGO1 function analysis [5]. Among them, *ago1-38*, which harbors an amino acid substitution in its N-coil domain, displayed peculiar morphological phenotypes, as described hereinafter (Figure 1A). In its juvenile stage, the *ago1-38* seedlings merely displayed mild growth defects in comparison to another commonly used *ago1-25* allele, which carried an amino acid substitution in the PIWI domain (Figure 1A,B). By contrast, the *ago1-45* allele, which harbors an amino acid substitution in its MID domain, also exhibited subtle growth defects (Figure 1A,B). However, as development progressed, the *ago1-38* allele displayed a series of distinctive abnormalities. One of the most obvious features was the pronounced upward growth of its shoot observed in 16-day-old plants (Figure 1B). As the plants reached maturity, the *ago1-38* allele exhibited peculiar and pleiotropic phenotypes, including severely curled and twisted leaves, small leaves regenerated from big ones, pocket-shaped leaves, slender petioles, severely delayed flowering, small siliques, and poor fertility (Figure 1C). Importantly, all these morphological defects in *ago1-38* were completely restored in genetic complementation lines harboring the *AGO1* promoter and its coding sequence, suggesting that its pleiotropic phenotypic defects directly resulted from the weakened AGO1 functions (Figure 1D). Collectively, these observations indicate that the mutation in the *ago1-38* allele compromises AGO1 activity and causes pleiotropic and peculiar morphological defects.

### 2.2. Metabolome Profiling of Three ago1 Mutants

The 21-day-old shoot tissues of *ago1-25*, *ago1-38,* and *ago1-45* were subjected to widely targeted metabolome analysis. A total of 2681 metabolites were identified in all samples (Appendix A), which were classified into 13 major categories (Figure 2A,B). Among them, alkaloids were the largest group, accounting for 14.32% of the total identified metabolites. Meanwhile, amino acids and their derivatives, flavonoids, lipids, and terpenoids each constituted more than 10% of the metabolite pool (Figure 2A). Principal component analysis (PCA) revealed less variations among the biological replicates of each genotype, but great variations among the WT, *ago1-25*, *ago1-38,* and *ago1-45* (Figure 2C), indicating pronounced metabolic differences among the three mutants. Thereafter, the differentially accumulated metabolites (DAMs) in *ago1-25*, *ago1-38,* and *ago1-45* in comparison to the WT were identified. A total of 430 DAMs (183 upregulated and 247 downregulated) in *ago1-38* (Figure 2D), 517 DAMs (333 upregulated and 184 downregulated) in *ago1-25* (Figure 2E), and 177 DAMs (86 upregulated and 91 downregulated) in *ago1-45* (Figure 2F) were detected. First, we analyzed the top 20 DAMs in each mutant, ranked by fold changes relative to the WT, and found that these top-scorning DAMs differed substantially among the three *ago1* alleles (Figure 2G–I). Next, we compared the distribution of the DAMs across the 13 aforementioned major metabolite classes (Figure 2A) and found distinct class-specific enrichment patterns for each mutant (Figure 2J–L). Finally, Kyoto Encyclopedia of Genes and Genomes (KEGG) analysis was performed to identify the metabolic pathways significantly changed in each allele using all the identified upregulated and downregulated DAMs. In *ago1-38,* significantly enriched pathways included the biosynthesis of caffeoylquinic acid derivatives, biosynthesis of kaempferol aglycones I, fructose and mannose metabolism, caffeine metabolism, and glutathione metabolism (Figure 2M). In comparison, the most enriched pathways in *ago1-25* included alpha-linolenic acid metabolism, fructose and mannose metabolism, linoleic acid metabolism, Purine metabolism, and Arginine and proline metabolism (Figure 2N). Meanwhile, *ago1-45* showed enrichment in pathways of anthocyanin biosynthesis, flavonoid biosynthesis, caffeine metabolism, and the biosynthesis of coumarins V (Figure 2O). Though a small number of pathways were simultaneously enriched in two or all three mutants, the most enriched pathways were allele-specific (Figure 2M–O). Taken together, these results suggest that each *ago1* variant triggers distinct metabolic reprogramming patterns.

In this study, we were particularly interested in the distinctive phenotype of *ago1-38*. Hence, we focused on metabolites specifically changed in *ago1-38* but not in *ago1-25* or *ago1-45*. A total of 286 *ago1-38*-specific DAMs were identified (Figure 3A). KEGG enrichment analysis showed that these 286 DAMs were significantly enriched in various pathways, including caffeoylquinic acid derivatives biosynthesis, other p-coumaric acid derivatives biosynthesis, caffeic acid derivatives biosynthesis, glucosinolate biosynthesis, ascorbate and aldarate metabolism, propanoate metabolism, and lysine degradation (Figure 3B). Meanwhile, we also analyzed the shared DAMs in three *ago1* mutants. Only 93 DAMs were shared between *ago1-25*, *ago1-38,* and *ago1-45* (Figure 3A) and KEGG analysis showed that these DAMs were significantly enriched in pathways including thyroid hormone synthesis, carbohydrate digestion and absorption, flavonoid biosynthesis, phosphotransferase system, flavone and flavonol biosynthesis, and anthocyanin biosynthesis (Appendix A). These results further indicate that *ago1-38* exhibits a metabolic profile distinct from *ago1-25* and *ago1-45*, although a subset of metabolic features is shared among the three alleles.

### 2.3. Transcriptomic Profiling of Three ago1 Mutants

Gene expression patterns serve as a molecular “blueprint” for phenotype formation [18]. To understand the potential mechanism underlying the distinct phenotype of *ago1-38*, transcriptional analysis was also employed. Shoots from 21-day-old WT, *ago1-25*, *ago1-38,* and *ago1-45* plants were harvested for transcriptome sequencing. The clear, high-quality reads were used for further analysis (Appendix A). PCA revealed less variations among biological replicates within each genotype, but great variations among the WT, *ago1-25*, *ago1-38,* and *ago1-45* (Figure 4A). Differentially expressed genes (DEGs) in *ago1-25*, *ago1-38,* and *ago1-45* compared to the WT were identified (Appendix A). A total of 1659 DEGs (839 upregulated and 820 downregulated) were identified for *ago1-38* (Figure 4B), representing the largest transcriptional shift among the three mutants. In comparison, 1324 DEGs (634 upregulated and 690 downregulated) were identified for *ago1-25* (Figure 4C), while 207 DEGs (90 upregulated and 117 downregulated) were identified for *ago1-45* (Figure 4D). KEGG enrichment analysis showed that the metabolic pathways were significantly enriched in all three mutants (Figure 4E–G), consistent with the broad role of AGO1 in regulating plant metabolism. However, plant–pathogen interactions, glutathione metabolism, and MAPK signaling pathways were significantly enriched in the *ago1-38* allele (Figure 4E), aligning with its distinct morphological and metabolic phenotypes. Meanwhile, GO enrichment analysis showed that pathways associated with plant responses to external stimuli and metabolic processes were significantly enriched in all three *ago1* mutants (Figure 4H–J), demonstrating the common roles of AGO1 in plant responses to external signal and plant metabolism. Notably, the pathway related to the plant response to salicylic acid was significantly enriched in the *ago1-38* allele (Figure 4H), suggesting a unique impact of *ago1-38* on defense hormone signaling. Take together, these results demonstrated that while all three *ago1* alleles share core transcriptional alterations, *ago1-38* displays distinct transcriptional reprogramming particularly linked to immune signaling pathways.

To compare the transcriptional reprogramming among the three alleles, we compared the most strongly enriched pathways identified by GO and KEGG analyses. GO enrichment analysis revealed significant differences in the pathways enriched across the three *ago1* mutants. In particular, *ago1-38* showed specific enrichment in pathways related to RNA-directed DNA polymerase, salicylic acid response, NAD(P)^+^ nucleotidase activity, glucosyltransferase activity, DNA polymerase activity, DNA integration, detoxification, and ADP binding (Figure 5A). KEGG analysis showed that the metabolic pathways and the biosynthesis of secondary metabolites were commonly enriched in all three *ago1* mutants (Figure 5B), consistent with the general regulatory role of AGO1 in metabolism. However, several pathways were specifically enriched in *ago1-38* but not in *ago1-25* or *ago1-45*, including those related to plant–pathogen interactions, zeatin biosynthesis, steroid biosynthesis, starch and sucrose metabolism, sesquiterpenoid and triterpenoid biosynthesis, arginine and proline metabolism, anthocyanin biosynthesis, and amino sugar and nucleotide sugar metabolism (Figure 5B). Notably, the plant–pathogen interaction pathway was prominently enriched in *ago1-38* (Figure 5B), suggesting a distinct influence on immune-related signaling. These pathway-level comparisons further illustrate that *ago1-38* reconfigures multiple metabolic and signaling modules distinct from *ago1-25* and *ago1-45*, suggesting allele-specific rewiring of regulatory networks.

Next, we focused on the DEGs specifically present in *ago1-38* but absent in *ago1-25* and *ago1-45*. A total of 1338 *ago1-38*-specific DEGs were identified (Figure 5C), a number higher than the set of DEGs shared among all three mutants. GO analysis showed that many of those *ago1-38*-specific DEGs were associated with phytohormone-related pathways, particularly those involving salicylic acid (SA), jasmonic acid (JA), and abscisic acid (ABA) (Figure 5D). Importantly, genes involved in the small RNA processing pathway as well as the RdDM pathway were also enriched (Figure 5D). Consistently, KEGG analysis also showed that the plant hormone signal transduction pathway as well as the plant–pathogen interaction pathway were also extremely significantly enriched in *ago1-38* (Figure 5E). Additionally, pathways including MAPK signaling, the ABC transporter, and the RNA polymerase function were also significantly enriched in *ago1-38* (Figure 5E). Most importantly, four genes related to the RNA polymerase were differentially expressed (Figure 5E), including nuclear DNA-dependent RNA polymerases Ⅳ (*NRPD4,* a subunit of RNA polymerase Ⅳ and V), *NRPB6B* (a subunit of RNA polymerase Ⅱ and lⅤ), *AT4G01590* (DNA-directed RNA polymerase III subunit), and *AT4G07950* (DNA-directed RNA polymerase, subunit M). Among these, *NRPD4* and *NRPB6B* were upregulated, while *AT4G01590* and *AT4G07950* were downregulated (Figure 5F). In contrast, the key components of Pol Ⅱ (*NRPB1* and *NRPB2*) were almost unchanged (Figure 5F), and *NRPD1* and *NRPE1*, the largest subunit of Pol Ⅳ and Pol Ⅴ, were merely slightly downregulated and upregulated, respectively (Figure 5F). In the siRNA processing pathway, five genes, *RDR3*, *RDR4*, *RDR5*, *NRPD4*, and *SGS3*, were enriched. *RDR3* and *RDR4* were significantly downregulated, while *RDR5*, *NRPD4*, and *SGS3* were significantly upregulated (Figure 5F). Other *RDR* and *SGS* family members, such as *RDR2*, *RDR1*, *RDR6* (also known as *SGS2*), and *SGS1,* showed no significant expression variation (Figure 5F). KEGG analysis further identified 95 genes involved in the plant–pathogen interaction pathway, with 29 genes upregulated and 66 genes downregulated, as well as 48 genes associated with the plant hormone signaling pathway, of which 32 were upregulated (Figure 5G). Meanwhile, GO analysis also showed that genes involved in SA, JA, and ABA signaling pathways were either up- or downregulated (Figure 5G). Collectively, these data suggested that pathways associated with siRNA processing, RdDM, plant–pathogen interactions, and phytohormone signals were significantly disturbed in *ago1-38*.

Finally, we analyzed 77 DEGs that were shared among the three *ago1* mutants (Figure 5C). KEGG analysis showed that these common DEGs were significantly enriched in pathways of cyanoamino acid metabolism, flavonoid biosynthesis, anthocyanin biosynthesis, and starch and sucrose metabolism (Appendix A). Consistently, GO analysis showed that these DEGs were significantly enriched in pathways of flavonoid biosynthetic, anthocyanin-containing compound metabolic, and glucosyltransferase activity (Appendix A). Together, KEGG and GO analyses further support the common role of AGO1 in plant metabolic pathways.

To identify gene expression patterns related to the *ago1-38* phenotype, weighted correlation network analysis (WGCNA) was conducted using DEGs from each *ago1* mutant, including all biological replicates. Gene clustering analysis identified 12 distinct co-expression modules, visualized as colored branches (Figure 6A). The relationships between gene module expression and different samples were assessed using correlation with a Pearson correlation coefficient in the dendrograms (Figure 6B). To infer potential biological functions of these modules, GO enrichment was conducted for each cluster. The brown, yellow, magenta, pink, black and greenyellow modules were significantly enriched in pathways related to small RNA processing and phytohormone signaling (Figure 6C). Meanwhile, the pink module contained 15 genes associated with the leaf development pathway (Figure 6C). Collectively, these data further indicate a relationship between *ago1-38* with dysregulation of the small RNA processing pathway and phytohormone signaling cascades.

### 2.4. The G186R Substitution in ago1-38 Does Not Affect Its Gene-Binding Activity

A previous study reported that AGO1 can directly bind to specific genes and activate their expression, particularly those involved in JA and SA signaling [10,11]. Our results also showed that *ago1-38*-specific DEGs were significantly enriched in phytohormone-related pathways, including JA and SA signaling (Figure 4H, Figure 5D,E and Figure 7B,C). To investigate whether G186R substitution affects AGO1’s binding to target genes, we compared the *ago1-38*-specific DEGs identified in this study with AGO1-binding genes previously reported. Notably, no overlapped genes were observed between *ago1-38*-specific DEGs and the previously identified AGO1-binding genes under both normal and stimuli conditions (Appendix A). These results indicated that the G186R substitution in *ago1-38* does not affect AGO1’s ability to bind its known target genes.

### 2.5. Expression of Differentially Expressed Transcription Factors in ago1-38

The above results showed that several pathways, including phytohormone signaling, were significantly disturbed in *ago1-38*. To further explore the transcriptional regulation underlying these changes, we analyzed the expression patterns of transcription factors (TFs) in the *ago1-38*-specific DEGs. Overall, 105 TFs were identified among the 430 DEGs in *ago1-38*, of which 77 were uniquely altered in this allele (Figure 7A). GO analysis showed that these 77 enriched TFs were extensively involved in various phytohormone response pathways, particularly those associated with auxin, ABA, SA, ethylene (ETH), cytokinin (CTK), JA, and gibberellic acid (GA) signaling (Figure 7B). Consistently, KEGG analysis showed that phytohormone signaling and MAPK signaling pathways were significantly enriched in the *ago1-38* allele (Figure 7C). Meanwhile, the circadian rhythm and plant–pathogen interaction pathways were also significantly enriched (Figure 7C). Most TFs related to these phytohormone signaling networks were significantly upregulated in *ago1-38* (Figure 7D). For comparison, 154 and 24 TFs were also identified among the DEGs of *ago1-25* and *ago1-45*, respectively (Figure 6A), indicating that transcriptional regulation was affected across all three *ago1* mutants. Take together, these results suggest that *ago1-38* causes extensive reprogramming of transcription factor expression, particularly those involved in phytohormone signaling cascades, which may underlie its distinctive and pleiotropic developmental phenotypes.

### 2.6. Integration of Transcriptome and Metabolome Profiles

To identify common biological processes in the three *ago1* alleles, we examined KEGG pathways that were simultaneously enriched by DEGs and DAMs. For each mutant, the 25 top-ranked co-enriched pathways were displayed. In *ago1-38*, only glutathione metabolism was the prominently co-enriched pathway (Figure 8C). Unfortunately, no pathways were significantly co-enriched in *ago1-38*-specific DAMs and DEGs (Figure 3A, Figure 5C and Figure 8D). This finding highlights a potential link between redox regulation and the pleiotropic developmental defects of *ago1-38*. For comparison, in *ago1-25*, only the alpha-linolenic acid metabolism pathway was prominently co-enriched (Figure 8A), whereas in *ago1-45*, the most co-enriched pathways were anthocyanin biosynthesis and flavonoid biosynthesis (Figure 8B). Collectively, these results further highlight the distinct metabolic and transcriptional reprogramming among *ago1-25*, *ago1-38,* and *ago1-45*, with glutathione metabolism specifically co-enriched in *ago1-38*.

## 3. Discussion

AGO1 is the first discovered AGO protein in Arabidopsis, and null mutants (e.g., *ago1-24* allele) of *AGO1* cause severe growth arrest and sterility. AGO1 associates with a large number of endogenous sRNAs to form a complicated regulatory network to suppress target genes involved in plant growth, development, and stress responses [19]. The main functions of AGO1 in plant growth and development include the determination of plant stature, leaf shape, flower phenotypes, sterility, adventitious rooting, and shoot apical meristem (SAM) development, and AGO1 is also required for plant resistant to various viruses and pathogens [19]. Except for its cannonical sRNA-binding role, nuclear AGO1 also binds to chromatin to activate some gene expressions under stimuli conditions, such as phytohormones JA and SA [10,11]. In Addition, rencent studies also show that the N-terminal domian of AGO1 is also very important for its activity. The N-coil acts as a structural switch to maintain AGO1’s stability in its RNA-free state through direct interaction with ATI1 in the autophage pathway [6]. Additionally, the N-terminal extension of AGO1 acts as an essential hub for PRMT5 interaction and post-translational modifications of symmetric arginine dimethylation in this domain [20]. In this study, we focused on *ago1-38*, which harbors amino acid substitution in its N-coil doamin and displays a unique morphological phenotype. We first performed the genotypic analysis of *ago1-38*; then, we performed the metabolomic and transcriptomic analysis of *ago1-38* independently; finally, the integrated metabolomic and transcriptomic analysis was performed to identify the co-enriched pathways in *ago1-38* (Figure 9A–C). Several important pathways were enriched in *ago1-38*, but it was difficult to clearly state whether these pathways were up- or downregulated, and the key indicators involved in these pathways still need to be identified. In addition, correction between the identified DAMs and DEGs and the *ago1-38* phenotype need to be clarified in the future.

### 3.1. The Peculiar Phenotype of ago1-38

In our laboratory, the commonly used *ago1* alleles are *ago1-25*, *ago1-27,* and *ago1-45*. The *ago1-38* allele carries an amino acid substitution of Gly with Arg (G186R), which is predicted to dislocate the N-terminal coil and distort its structural stability [21,22]. Although *ago1-25* and *ago1-45* also displayed pleiotropic growth defects, similar phenotypes could be observed in other common mutants (e.g., *dcl1* and *hyl1* in miRNA biogenesis-related pathway) in the sRNA-related pathways. However, the *ago1-38* mutant displayed a unique phenotype that is rarely seen in other mutants. The most distinctive phenotypes of *ago1-38* include the following: ultra-twisted and curled leaves, slender petioles, asymmetric leaves, small leaves regenerated from large ones, and also pocket-shaped leaves (Figure 1C,D). Furthermore, *ago1-38* also exhibited severely delayed flowering, short and small siliques, and poor fertility; all of those are phenotypically more pronounced than what was observed in *ago1-25* and *ago1-45* (Figure 1C,D). All of these developmental defects were completely restored in their complementary alleles, suggesting that the pleiotropic phenotype of *ago1-38* resulted from the reduced function of AGO1. Regarding why we selected *ago1-38* but not the other alleles for in-depth analysis, there are two reasons that cannot be ignored: (i) analyzing the “rare” phenotype displayed by *ago1-38* may lead to the uncovering of new functions of AGO1; (ii) mutation in the N-coil domain of AGO1 is also “rare”, analyzing this mutant may also lead to the finding of new functions of the N-coil domain of AGO1.

### 3.2. The Distinct Metabolic Profile of ago1-38

The metabolic profile and plant phenotype are usually close correlated, with the metabolic profile being a crucial component of the plant phenotype [23], which offers a predictive tool to identify metabolic markers associated with plant performance to enable accelerated crop improvement [24]. In this study, a total of 517, 430, and 177 DAMs were identified in *ago1-25*, *ago1-38,* and *ago1-45*, respectively (Figure 2D–F). Coincidently, the number of DAMs is roughly correlated with the overall severity of phenotypic alterations. Among these three *ago1* mutants, *ago1-25* displayed the most severe overall phenotype, whereas *ago1-45* displayed the mildest phenotype at the mature plant stage (Figure 1C). Although *ago1-38* displayed only moderate vegetative defects, its polymorphic leaf shape and severely reduced fertility distinguish it from the other alleles (Figure 1C). Meanwhile, the DAMs identified in *ago1-25*, *ago1-38,* and *ago1-45* are merely poorly overlapped, and only 93 DAMs are shared among these mutants (Figure 3A). By contrast, there are 336 *ago1-25*-specific, 286 *ago1-38*-specific, and 49 *ago1-45*-specific DAMs identified (Figure 3A). These results suggest the markedly different patterns of metabolic regulation in these mutants. Furthermore, the enriched metabolic pathways in these mutants were also largely allele-specific, with *ago1-38* showing a characteristic enrichment in the glutathione metabolism pathway (Figure 2M and Figure 8C)—which is closely associated with redox homeostasis and stress adaptation. The distinct metabolic pattern suggests that *ago1-38* undergoes specialized metabolic reprogramming that may contribute to its unique developmental and fertility phenotypes.

### 3.3. The Distinct Transcriptomic Profile of ago1-38

The transcriptomic profile and plant phenotype are tightly corrected. Phenotype is a direct manifestation of gene function, while transcriptome data can reveal the activity state of genes in specific environments or developmental stages, thereby helping to analyze the molecular mechanisms underlying phenotype [25,26]. Like the case observed in metabolic profiles, the transcriptomic profiles of *ago1-25*, *ago1-38,* and *ago1-45* are also diversely regulated. There are 944 *ago1-25*-specific, 1338 *ago1-38*-specific, and 19 *ago1-45*-specific DEGs identified (Figure 3A). However, only 77 common genes are shared among these three alleles (Figure 3A). Expectedly, only 19 genes are specifically regulated in *ago1-45*, which displayed the weakest phenotype. Nevertheless, up to 1338 genes are specifically regulated in *ago1-38* (Figure 3A), which is a rather larger number than *ago1-25*. The ago1-38 allele displayed a moderate but complex phenotype. Thus, the number of DEGs in *ago1-38* may be correlated with the complexity of its phenotype. The distinct transcriptomic pattern suggests that *ago1-38* also undergoes specialized transcriptomic reprogramming that may contribute to its complicated developmental and fertility phenotypes.

### 3.4. The Dysregulation of siRNA Processing Pathway in ago1-38

The canonical function of AGO1 is to load sRNAs and mediate post-transcriptional gene silencing. In this study, we observed significant transcriptional changes in several genes involved in siRNA processing in *ago1-38* (Figure 5D–F). These genes can mainly be categorized into three major classes. The first class includes genes involved in encoding subunits of RNA polymerase, particularly, Pol Ⅳ and Pol Ⅴ were possessed specifically by plants [27,28]. The second class comprises genes that encode RDR and SGS proteins, which are responsible for the formation of double-strand RNA prior to DICER-mediated cleavage [29,30]. The third class consists of genes functioning in the RdDM pathway [31,32], which is responsible for transcriptional gene silencing (TGS). These transcriptional perturbations suggest that G186R substitution in *ago1-38* not only weakens AGO1’s canonical post-transcriptional silencing activity but may also influence upstream components of siRNA biogenesis and also DNA methylation at a transcriptional level, thereby revealing the unconventional regulatory roles of AGO1 in siRNA processing and DNA methylation.

### 3.5. The Disturbed Plant–Pathogen Interaction Networks in ago1-38

KEGG analysis showed that the plant–pathogen interaction pathway was significantly enriched in *ago1-38*-specific DEGs (Figure 4E and Figure 5B,E) and also in *ago1-38*-specific transcriptional factors (Figure 7C). SiRNA-mediated pathogen defense has been widely reported in a previous study [33], and the concurrent enrichment of pathogen-defense and siRNA processing pathways in *ago1-38* suggests the possible interaction between these two pathways. Nevertheless, establishing the causal relationship in *ago1-38* between these two pathways remains challenging.

### 3.6. The Disturbed Phytohormone Signaling Cascades in ago1-38

Phytohormones are key signaling molecules that regulate plant growth, development, and responses to the environment, ultimately shaping the plant’s phenotype [34]. Multiple methods of analysis, including GO, KEGG, and WGCNA, consistently showed that the phytohormone signaling pathways were significantly enriched in *ago1-38*-specific DEGs (Figure 5A,B,D,E,G and Figure 8C), as well as in *ago1-38*-specific TFs (Figure 7B–D). The massive phytohormone signaling-related genes, including a large number of TFs, exhibited both up- and downregulated expression patterns in *ago1-38*, suggesting evidence of rather complicated and disturbed phytohormone signaling in *ago1-38*. However, whether the peculiar morphological and fertility phenotype observed in *ago1-38* directly resulted from the perturbed phytohormone signaling remains to be investigated in future studies. Additionally, although a large number of genes in *ago1-38* were differently expressed, these genes were not overlapped with the genes identified in previous study that were specifically bound and activated by AGO1 (Appendix A). Hence, how these genes in phytohormone signaling are specifically regulated in *ago1-38* still needs further research.

## 4. Materials and Methods

### 4.1. Plant Cultivation and Growth Conditions

*Arabidopsis* thaliana wild-type Columbia-0 (Col-0) and three point mutants *ago1-25*, *ago1-38,* and *ago1-45*, each harboring an amino acid substitution of AGO1 in the Col-0 background were used in this study. The *ago1-25*, *ago1-38,* and *ago1-45* mutants were obtained from the mutant collection maintained in our laboratory. Seeds were surface-sterilized with 75% ethanol for 10 min and then sown on 1/2 MS medium consisting of 2.2 g/L Murashige and Skoog basal salt mixture (Phyto Tech Labs, Cat. No. M524, Lenexa, KS, USA), 10 g/L sucrose, and 10 g/L agar. The pH was adjusted to 5.8 with 5 M NaOH before autoclaving. After sowing the seeds, the Petri dishes were sealed with Micropore Scotch 3 M surgical tape to prevent contamination while allowing for a gaseous exchange and then were placed in a 4 °C cold room for 2 days in the dark for vernalization. Seedlings were grown at 22 °C in Percival tissue culture chambers under long-day conditions (16 h light/8 h dark). At approximately 10-days-old, seedlings were transplanted into soil pots containing a 0.5:3:9 (*v*/*v*/*v*) mixture of perlite, soil, and vermiculite at 22 °C under the same long-day photoperiod illuminated a combination of incandescent and fluorescent lamps (10,000 lux). Plants were watered twice a week with nutrient solution. Shoots from 21-day-old plants were harvested for total RNA and metabolite extraction. Three biological replicates were prepared for each genotype for both transcriptomic and metabolomic analyses.

### 4.2. Construction of ago1-38 Complementation Lines

A DNA fragment harboring a 2000 bp promoter region and an intact AGO open reading frame was cloned into a modified pMDC107 vector via homologous recombination; the resulting construct was then introduced into the *ago1-38* mutant by Agrobacterium-mediated genetic transformation. Positive transformants were screened on 1/2 MS medium supplemented with 50 ng/μL hygromycin and confirmed by PCR analysis using genomic DNA as template.

### 4.3. Widely Targeted Metabolome Analysis

Shoot samples of 21-day-old WT, *ago1-25*, *ago1-38,* and *ago1-45* plants with three biological replicates were harvested (each replicate including 5–10 g shoot sample collected form more than 20 plants) and then immediately froze by liquid nitrogen to stop metabolism, samples were then transferred to dry ice and sent to Metware Biotechnology Co., Ltd. (Wuhan, China) for metabonomic analysis. Samples were grounded into fine power using liquid nitrogen and then placed in a 70% aqueous methanol. After centrifugation at 10,000× *g* for 10 min, all supernatants were combined and filtered through a 0.22 μm pore size membrane and then analyzed using a UPLC-ESI-MS/MS system (UPLC, ExionLC™ AD, SCIEX, Framingham, MA, USA, https://sciex.com.cn/) and Tandem mass spectrometry system (https://sciex.com.cn/). The reference standard was used for quality control. Based on the detected metabolites, principal component analysis (PCA) was performed with FactoMineR and factoextra packages in R to visualize clustering patterns among the samples. Orthogonal partial least squares discriminant analysis was performed to determine the DAMs using a threshold of |log2FoldChange| > 1 and variable importance in project ≥ 1. Based on the metware database, identified metabolites were annotated using the newly updated KEGG compound database, and mapped to the KEGG pathways.

### 4.4. Transcriptome Analysis

The samples were collected the same way as in metabolome analysis. Total RNA was extracted from the shoots of 21-day-old WT, *ago1-25*, *ago1-38,* and *ago1-45* plants using an RNA extraction kit (Tiangen Biotech Co., Ltd., Cat. No. DP432, Beijing, China). RNA quantity was subsequently evaluated with a Nanodrop One Ultra-Micro Spectrophotometer (Thermo Fisher Scientific, Waltham, MA, USA), and RNA quality was further verified using agarose gel electrophoresis. RNA samples that met the quality standards were kept in dry ice and sent to Metware Biotechnology Co., Ltd. (Wuhan, China) for cDNA library construction and sequencing on the BGISEQ sequencing platform. Raw data (raw reads) in FASTQ format were initially processed using Trimmomatic (v0.39) with the following parameters: ILLUMINACLIP: adapters.fa:2:30:10, LEADING:3, TRAILING:3, SLIDINGWINDOW:4:20, and MINLEN:36. During this step, clean data (clean reads) were obtained by removing adapter sequences, reads containing poly-N sequences, and low-quality reads. In addition, the Q20, Q30, and GC contents of the clean data were calculated. All downstream analyses were conducted using these high-quality clean data. Gene expression levels were quantified using FPKM (Fragments Per Kilobase of transcript per Million mapped reads). Differentially expressed genes (DEGs) between genotypes were identified with DESeq2 with thresholds of |log2(fold change)| ≥ 1 and FDR < 0.05. Identified DEGs were annotated using the newly updated KEGG and GO database, and mapped to the KEGG and GO pathways, respectively.

### 4.5. Integrated Transcriptomic and Metabolomic Analysis

To integrate the transcriptomic and metabolomic datasets, KEGG pathway analysis was conducted to identify pathways co-enriched by DEGs and DAMs. Pathway enrichment analysis was performed and pathways with *p* values of *p* < 0.05 were considered significantly enriched pathways for both genes and metabolites. The KEGG enrichment results were visualized using the ggplot2 package in R software (Version 4.5.1; R Core Team, 2024), and the major metabolic pathways were plotted according to the KEGG database.

## 5. Conclusions

The *ago1-38* allele displayed a peculiar phenotype distinct from the other commonly used *ago1* mutants. Our results showed that *ago1-38* also displayed distinct transcriptomic and metabolomic profiling in comparison to *ago1-25* and *ago1-45*. The glutathione metabolism pathway was specifically co-enriched in *ago1-38* in integrated transcriptomic and metabolomic analysis, suggesting a distinctive role of AGO1 in maintaining redox homeostasis and stress adaptation. The expression of a subset of genes in the siRNA processing pathway, including the encoding components of Pol Ⅳ and Pol Ⅴ, RDR and SGS, and players in the RdDM pathway, was significantly disturbed in *ago1-38*, suggesting a role of AGO1 in siRNA generation and function. The plant–pathogen interaction pathway was also significantly enriched in *ago1-38*, which may be linked with simultaneously changed genes related to the siRNA processing pathway. The phytohormone signaling pathway was also significantly enriched in *ago1-38*, which may tightly associate with the peculiar phenotype of *ago1-38*. Given that the siRNA processing pathway, pathogen response networks, and phytohormone signal cascades are usually intricately interconnected, with each pathway affecting the others, our findings systematically revealed the distinct metabolic and transcriptomic profiling of *ago1-38* and provided new insights into the multifaceted roles of AGO1 in coordinating siRNA processing, pathogen defense, and phytohormone signaling, thereby modulating plant development and phenotype.

## Figures and Tables

**Figure 1 plants-15-00044-f001:**
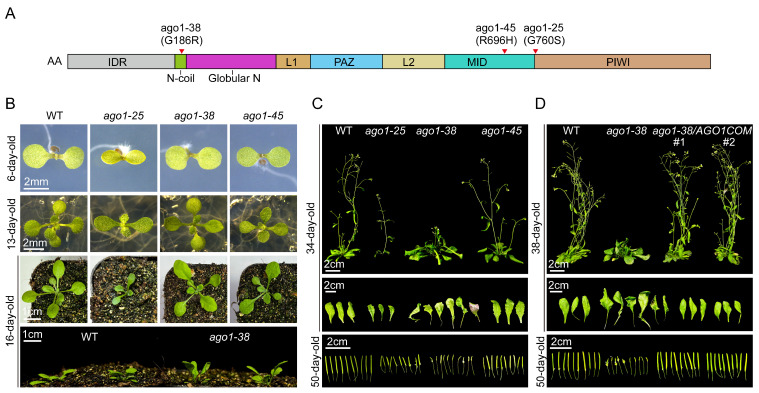
**The peculiar phenotype of the *ago1-38* allele.** (**A**) Schematic representation of the domain structure of AGO1 slicer protein, referenced from a previous report [13]. Individual domains were marked by different colors. The positions of the three missense alleles were depicted by red triangles. The *ago1-38* allele harboring a G186R substitution in N-coil domain, the *ago1-25* allele harboring a G760S substitution in MID domain, and the *ago1-45* allele harboring an R696H substitution in PIWI domain in protein levels. (**B**) Representative shoot phenotypes of 6-day-old seedlings and 13-day-old young plants grown on 1/2MS medium, and 16-day-old small plants grown in soil for *ago1-25*, *ago1-38,* and *ago1-45* alleles. (**C**) Morphological phenotypes of 34-day-old mature plants, detached rosette leaves from the same plants, and siliques detached from 50-day-old *ago1-25*, *ago1-38,* and *ago1-45* plants. (**D**) Morphological phenotypes of 38-day-old *ago1-38* and *ago1-38*/COM transgenic complementation lines, detached rosette leaves from the same plants, and siliques from 50-day-old *ago1-38* and *ago1-38*/COM transgenic plants, the two independent lines were labeled as #1 and #2 for distinction. In (**B**–**D**), scale bars are indicated in each panel.

**Figure 2 plants-15-00044-f002:**
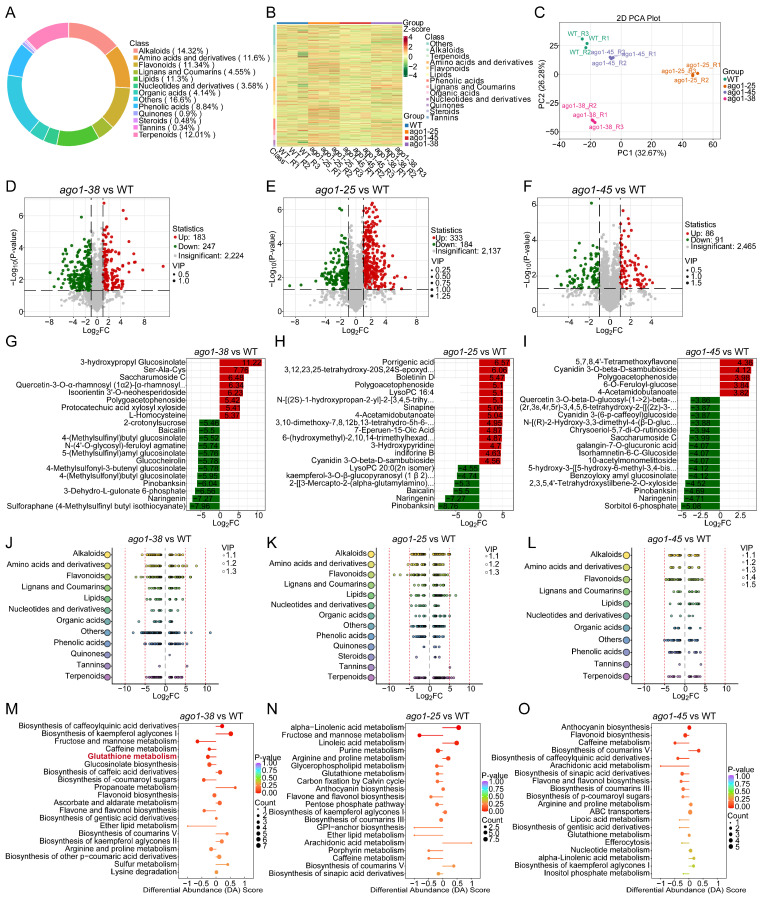
**Classification of metabolites and identification of differentially accumulated metabolites.** (**A**) Ring diagram showing the classification and composition of identified metabolite category. (**B**) Heatmap clustering of identified metabolites. (**C**) Principal component analysis (PCA) of the metabolite profiles from WT, *ago1-25*, *ago1-38*, and *ago1-45* samples based on the identified metabolites. (**D**–**F**) Volcano plots showing the numbers of DAMs identified in pairwise comparisons of *ago1-38*, *ago1-25,* and *ago1-45* versus WT. Red and green dots represent up- and downregulated DAMs, respectively. (**G**–**I**) The top 20 DAMs in *ago1-38*, *ago1-25,* and *ago1-45* ranked by log_2_ fold change (log_2_FC). (**J**–**L**) Variable importance in projection (VIP) bar plots illustrating changes in metabolite profiles in *ago1-38*, *ago1-25,* and *ago1-45*. The *y*-axis represents compound classes, whereas the *x*-axis denotes log_2_FC values; bubble size corresponds to VIP scores. (**M**–**O**) Kyoto Encyclopedia of Genes and Genomes (KEGG) pathway enrichment and differential abundance (DA) score diagram of DAMs in *ago1-38*, *ago1-25,* and *ago1-45*. The bubble size represents number of DAMs, the color corresponds to *p*-value. In (**M**), the glutathione metabolism pathway was marked in red font, which was extensively enriched in henceforth analysis. The calculation of DA score in an enriched pathway: DA = (number of upregulated DAMs − number of downregulated DAMs)/number of total DAMs.

**Figure 3 plants-15-00044-f003:**
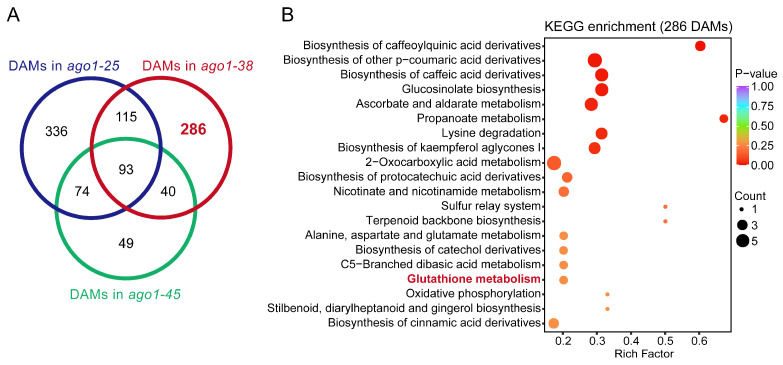
**Analysis of *ago1-38*-specific DAMs.** (**A**) Venn diagram showing the overlap of DAMs between *ago1-25*, *ago1-38,* and *ago1-45*. Each circle represents a comparison group. The number of the *ago1-38*-specific DAMs were marked in red bold font. (**B**) KEGG functional annotation and enrichment analysis of the 286 specific metabolites in *ago1-38* identified in (**A**). The glutathione metabolism pathway was marked in red font, which is extensively enriched in henceforth analysis. The *x*-axis represents the enrichment factor and the *y*-axis lists pathway names (sorted by *p*-value). Point color reflects *p*-value significance, with red indicating the highest and purple the lowest significance, while point size represents the number of differential metabolites.

**Figure 4 plants-15-00044-f004:**
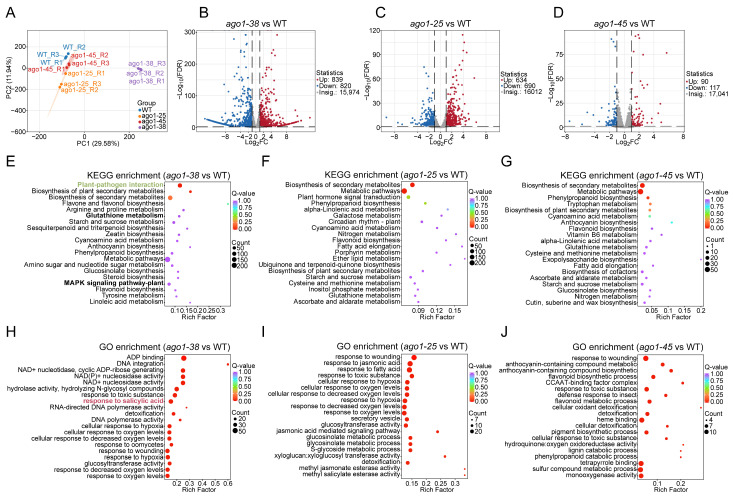
**Identification and functional enrichment analysis of DEGs.** (**A**) PCA of transcriptomic profiles from WT, *ago1-38*, *ago1-25*, and *ago1-45*. (**B**–**D**) Volcano plots showing differentially expressed genes (DEGs) in *ago1-38*, *ago1-25,* and *ago1-45* relative to WT. Red and blue dots represent upregulated and downregulated DEGs, respectively. (**E**–**G**) KEGG enrichment analysis of DEGs in *ago1-38*, *ago1-25,* and *ago1-45*. (**H**–**J**) Gene Ontology (GO) enrichment analysis of DEGs in *ago1-38*, *ago1-25,* and *ago1-45*. In (**E**,**H**), the plant–pathogen reaction pathway was marked in green, the glutathione metabolism and MAPK signaling pathways were marked in bold font, and the salicylic acid response was marked in red font; these pathways are also extensively enriched in henceforth analysis.

**Figure 5 plants-15-00044-f005:**
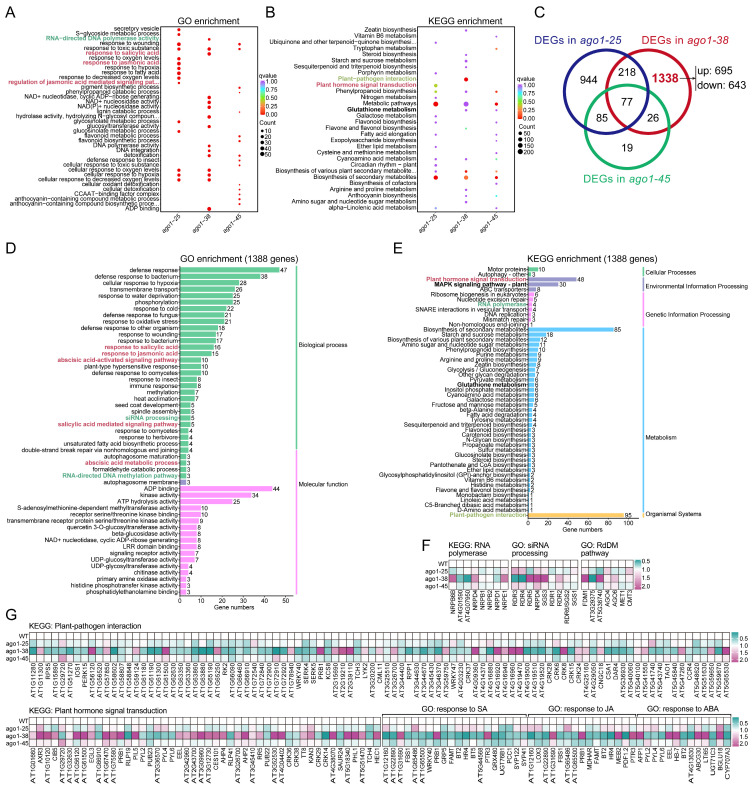
**In-depth analysis of genes specifically regulated in *ago1-38*.** (**A**) Comparison of enriched pathways in *ago1-38*, *ago1-25,* and *ago1-45* based on GO analysis. The phytohormone signaling-related pathways are marked in red font. (**B**) Comparison of enriched pathways in *ago1-38*, *ago1-25,* and *ago1-45* based on KEGG analysis. The phytohormone-related pathways are marked in red font. (**C**) Venn diagram of DEGs between *ago1-38*, *ago1-25*, and *ago1-45*. Each circle represents a comparison group. The 1338 genes specifically expressed in *ago1-38* are marked in red and bond font. (**D**) GO annotation and functional analysis of the 1338 *ago1-38*-specific DEGs. Phytohormone signaling-related pathways are marked in red font. The siRNA processing pathway is marked in green font. (**E**) KEGG annotation and functional analysis of the 1338 *ago1-38*-specific DEGs. Phytohormone signaling-related pathways are marked in red font. The RNA polymerase pathway is marked in green font. (**F**) Heatmap showing expression profile of phytohormone signaling-related DEGs based on FPKM values. Of note, some genes are shared in different hormones’ signaling. (**G**) Heatmap showing expression profiles of siRNA processing- and RNA polymerase-related DEGs based on FPKM values. Note that some genes are shared in different hormone signaling pathways. In (**A**,**B**,**D**,**E**), the siRNA, phytohormone, and plant–pathogen interaction pathways are marked in colored fonts.

**Figure 6 plants-15-00044-f006:**
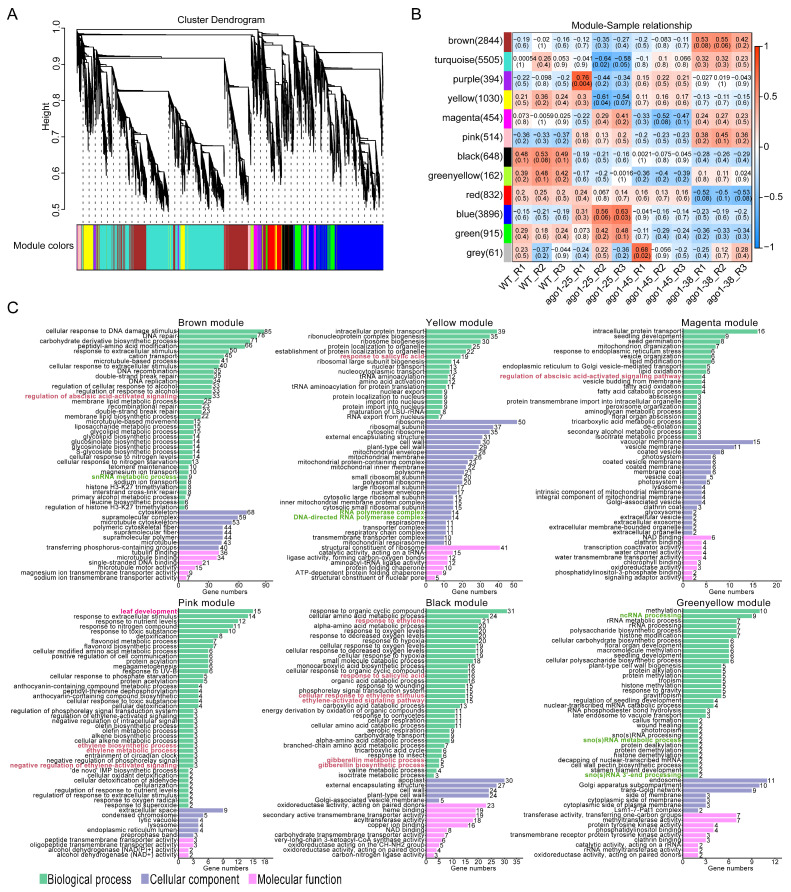
**WGCNA of transcriptomes identifies co-expression modules associated with sRNA processing and phytohormone in *ago1* mutants.** (**A**) Clustered modules generated from the weighted correlation network analysis. Each branch of the dendrogram represents genes, and genes clustered into the same module are assigned the same module color. Genes assigned in gray indicate no clustering to any module. (**B**) Module–trait relationship heatmap showing Pearson correlation coefficients between gene co-expression modules and sample traits. Each cell displays the correlation value and its *p*-value (red indicates positive correlations and blue indicates negative correlations, according to the color scale shown on the right). (**C**) GO analysis of genes clustered in brown module, yellow module, magenta module, pink module, black module, and greenyellow module. The phytohormone signaling-related pathways are marked in red font, the small RNA processing-related pathways are marked in green font, and the leaf development-related pathway is marked in pink font. The phytohormone signal, siRNA processing and leaf development pathways are highlighted in colored fonts.

**Figure 7 plants-15-00044-f007:**
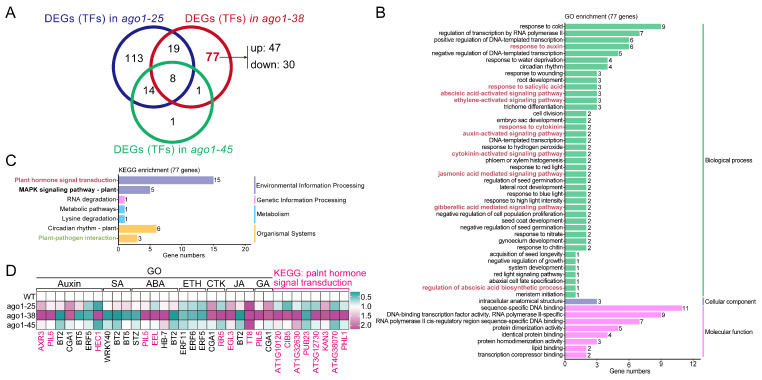
**In-depth analysis of transcriptional factors (TFs) specifically expressed in *ago1-38*.** (**A**) Venn diagram showing differently expressed TFs in *ago1-38*, *ago1-25,* and *ago1-45*. Each circle represents a comparison group. The 77 *ago1-38*-specific TFs are displayed in red font. (**B**) GO enrichment analysis of the 77 *ago1-38*-specific TFs. The phytohormone signaling-related pathways are marked in red font. (**C**) KEGG analysis of the 77 *ago1-38*-specific TFs. The phytohormone signaling-related pathway is marked in red font. (**D**) Heatmap showing expression profiles of genes involved in phytohormone signaling based on FPKM values. Some genes were co-enriched in GO analysis and KEGG analysis, and genes enriched in KEGG analysis are marked in pink font. Note that some genes are shared in different hormone signaling pathways. In (**B**,**C**), the phytohormone and plant–pathogen interaction pathways are marked in colored font; the MAPK signal pathway is marked in bold font.

**Figure 8 plants-15-00044-f008:**
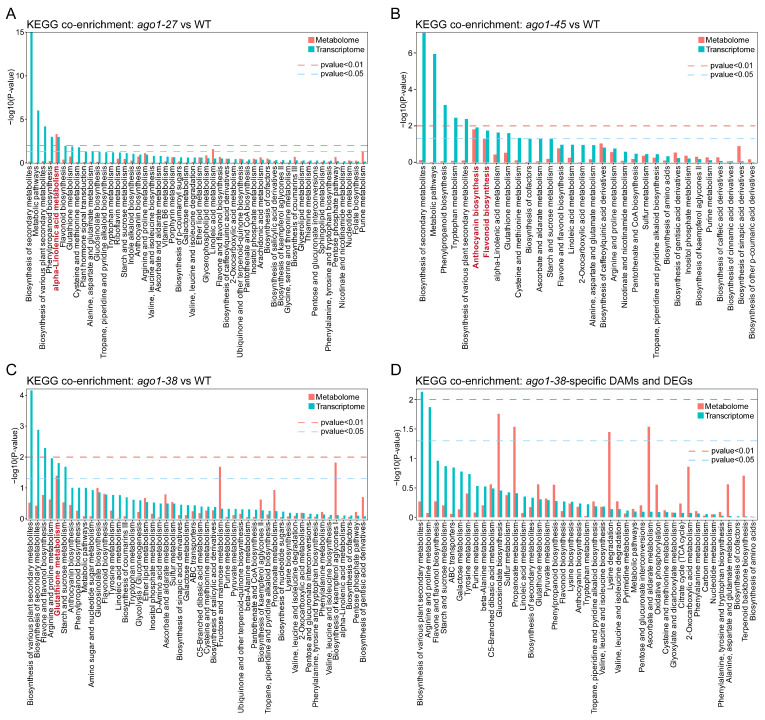
**Integrated analysis of transcriptome and metabolome.** (**A**) The co-enriched pathways in integrated KEGG analysis of DEGs and DAMs in the *ago1-25* allele. (**B**) The co-enriched pathways in integrated KEGG analysis of DEGs and DAMs in the *ago1-45* allele. (**C**) KEGG pathways co-enriched by DEGs and DEMs in *ago1-38* allele. (**D**) The co-enriched pathways in integrated KEGG analysis of *ago1-38*-specific DEGs and DAMs. The significantly co-enriched pathways in each allele were marked in red bold font.

**Figure 9 plants-15-00044-f009:**
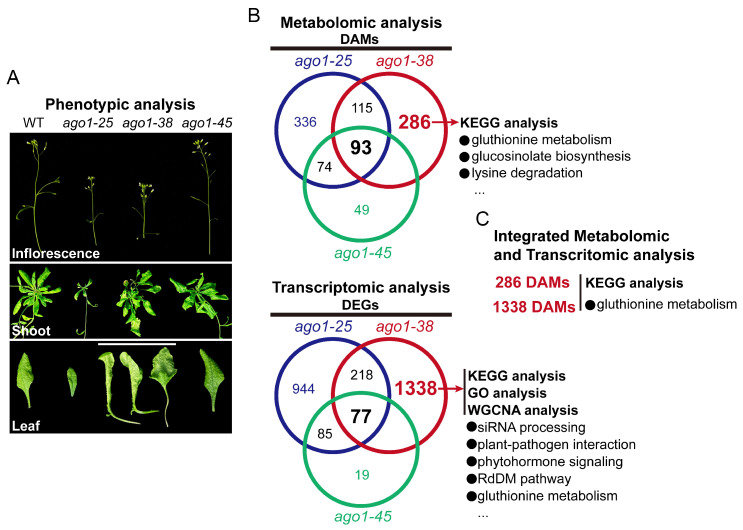
**A working model for the integrated metabolomic and transcriptomic analysis of *ago1-38*.** (**A**) Phenotypic analysis of *ago1-25*, *ago1-38,* and *ago1-45*. (**B**) Metabolomic analysis and transcritomic analysis of *ago1-25*, *ago1-38,* and *ago1-45*, respectively. The numbers of *ago1-38*-specific DAMs and DEGs were highlighted in red bold font. (**C**) Integrated metabolomic and transcritomic analysis of *ago1-38*.

## Data Availability

The transcriptomic data in fastq format used in the study can be found in the database (https://www.ncbi.nlm.nih.gov/bioproject/PRJNA1359165, accessed on 12 November 2025); the compound data obtained from metabolomic analysis can be found in Appendix A.

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
