# Peer review of "Integrative Transcriptomic and Metabolomic Analysis Provides New Insights into the Multifunctional ARGONAUTE 1 Through an Arabidopsis ago1-38 Mutant with Pleiotropic Growth Defects"

_plants, 2025, doi:10.3390/plants15010044_

Round 1

Reviewer 1 Report

Comments and Suggestions for Authors

This study systematically investigated the phenotypes of three Arabidopsis AGO1 mutants, with a particular focus on ago1-38, and their underlying molecular mechanisms through integrated transcriptomic and metabolomic analyses. This findings clearly elucidated the distinct phenotypes of Ago1-38 and revealed the associated molecular mechanisms. This research offers novel insights into the multifaceted roles of Ago1 in plant growth and development. Overall, this manuscript is well-structured with clear organization, appropriate figures and tables, and sound result analyses. The biggest problem with this manuscript is that the discussion is brief and superficial.

  1. Further strengthen the discussion of recent research progress on AGO1. Simplify the description of multi-omics technologies, as they are merely research methodologies.
  2. Incorporating more physiological and biochemical indicators for the three ago1mutants, and establishing their relationship with key DAMs and DEGs, would significantly enhance the manuscript's quality.
  3. The manuscript lacks validation experiments to confirm the accuracy of the metabolomic and transcriptomic results.
  4. Specify the timeline for database comparisons of transcriptomic and metabolomic data.
  5. Use standardized descriptions for reagents and materials, e.g., Line 524, RNA extraction kit (Tiangen).
  6. Line 112, please revise the title.
  7. Remove Chinese characters from Table S1.
  8. Irecommend the authors construct a diagram illustrating the proposed mechanism of action.
  9. The discussion section requires substantial strengthening. The current draft merely summarizes the study's findings and lacks comparative analysis with previous research, as well as in-depth interpretation. It should also address the limitations of the current study (e.g., lack of mechanistic validation) and propose future research directions.
  10. Carefully revise the format of the references and add more references. The fact that there are only two references in the discussion section also reflects the inadequacy of the discussion.

Reviewer 2 Report

Comments and Suggestions for Authors

Review of manuscript plants-4017593 entitled “Integrative Transcriptomic and Metabolomic Analysis Provides New Insights into the Multifunctional ARGONAUTE 1 Through an Arabidopsis ago1-38 Mutant with Pleiotropic Growth Defects”

Summary

The work presents a multi-omics characterization of the Arabidopsis ago1-38 mutant, which shows peculiar phenotypes compared with the ago1-25 and ago1-45 alleles. The integrated analysis of the two omics layers identifies 1,338 genes and 286 metabolites specifically altered in ago1-38. The most affected pathways include glutathione metabolism, siRNA processing, hormonal signaling, and plant–pathogen interaction. These alterations suggest a multifaceted impact of the mutation on redox regulation, defense, and development.

Commentaries

In my opinion, the authors present a solid and well-organized study that offers a comprehensive multi-omics characterization of the Arabidopsis ago1-38 mutant compared with other ago1 alleles. The experimental approach is appropriate, and the datasets are clearly explained, providing useful insight into allele-specific metabolic and transcriptional changes. In summary, the manuscript fits well within the scope of Plants. However, I recommend that the authors clarify a few points and moderate the strength of some statements before the manuscript can be accepted.

Abstract

It is too long. The first part reads like an introduction. It should be completely rewritten to reduce the length and focus on the results. The abstract should also concentrate on statements that are fully supported by the data.

Introduction

In general, the introduction is appropriate. However, I think the authors should justify more clearly why the ago1-38 allele was selected for detailed analysis, and why ago1-25 and ago1-45 were chosen as the control alleles. Could this specific choice introduce bias in the interpretation of allele-specific effects?

Methods

Metabolomics

There are three biological replicates, but how many plants/leaves were included in each?

What treatment was applied to the shoots immediately after harvest? I am not sure if this should be described here or in the previous section. Were the samples frozen to stop metabolism? How were they preserved?

It would be helpful to provide more detail on the experimental protocol, even in the supplementary material.

Were QC samples prepared?

More information is needed to understand how the metabolite list used for downstream analysis was obtained.

In addition, the information on the statistical analysis is very limited, which makes it difficult to follow the workflow. Please expand.

Transcriptomics

Please expand the description of the library preparation and the read type.

As above, more detail on the statistical parameters used would be appreciated.

QCs again?

Multi-omics

With the datasets generated, a more complete multi-omics analysis could be performed to minimize false positives. Please give more detail on how KEGG was used and how this supports the reliability of the results.

Results

2.1 Ago1-38 allele

The text is a bit long, and I think it would be good to reduce some of the descriptive detail. However, it should be slightly expanded by adding quantitative measurements that allow the reader to clearly see how ago1-38 differs from the other variants.

2.2. Metabolomic profiling

How are the metabolites characterized in the targeted analysis? Please provide more detail either here or in the Methods section.

Figure 2 contains too much information, which makes all elements appear too small. I recommend splitting the figure into several panels and, in some cases, moving parts of it to the supplementary material.

How are the differences in the number of DAMs justified? And why are the top 20 selected? It would be preferable to use a threshold based on fold change (up or down) or statistical significance.

Line 165: How many metabolites were included in the KEGG analysis

2.3 Transcriptomics

Figure 4/5 (same comment as for Figure 2).

I miss information on fold changes, which would help identify which genes show the strongest differences (this also applies to the previous section).

2.4 G186R mutation

Which genes were selected for the comparison? Is this information available in the supplementary material? How can we be sure that the genes being compared come from equivalent conditions? If this cannot be guaranteed, the statement should be moderated.

2.6 WGCNA

I think this section could be shortened and moved into the transcriptomics results, as it leads to conclusions similar to those already described.

2.7 Multi-omics

I do not think that simply searching for overlaps between DEG and DAM lists can be considered a multi-omics analysis. The authors should attempt to identify correlations using more robust integration methods.

As mentioned before, the choice of the “top 25” pathways seems arbitrary and may be the reason why the overlap is limited. The analysis method should be improved.

Discussion

In general, this section discusses the results only superficially and focuses more on restating them. I believe it may be better to restructure the manuscript as a combined Results and Discussion section, or alternatively to rewrite the discussion with greater depth.

In addition, based on the metabolomic and transcriptomic results and the pathways affected, some experimental validation (gene expression by qPCR, ROS, GSH, etc.) could be performed to support certain statements (e.g., redox effects, siRNA-related conclusions).

Furthermore, the comparison with other AGO1 alleles should be explored in greater depth using the existing literature. If I am not mistaken, there are only two citations in this entire section, which does not strongly support the authors’ claims.

Round 2

Reviewer 1 Report

Comments and Suggestions for Authors

The author has responded to the comments and revised the manuscript, which is a very good research paper. There is a suggestion for modification, as stated in the author's response, the biological database is updated quickly, so the author needs to indicate the comparison time of the database in the manuscript. For example, KEGG (http://www.kegg.jp/kegg/pathway.html, accessed on 10 July 2025).

Reviewer 2 Report

Comments and Suggestions for Authors

Review of manuscript plants-4017593 entitled “Integrative Transcriptomic and Metabolomic Analysis Provides New Insights into the Multifunctional ARGONAUTE 1 Through an Arabidopsis ago1-38 Mutant with Pleiotropic Growth Defects“

The authors have successfully addressed the reviewer comments. The manuscript is accepted for publication, subject to a final check for typographical and punctuation errors.

Author Response

Thank you very much for taking the time to review this manuscript.